

**Evaluating the sensitivity of fine particulate matter (PM$_{2.5}$) simulations to chemical**
**mechanism in Delhi**
Chinmay Jena[1*], Sachin D. Ghude[1*], Rachana Kulkarni[1], Sreyashi Debnath[1], Rajesh Kumar[2], Vijay Kumar
Soni[3], Prodip Acharja[1], Santosh H. Kulkarni[4], Manoj Khare[4], Akshara J. Kaginalkar[4], Dilip M. Chate[4], Kaushar
Ali[1], Ravi S. Nanjundiah[1,5], Madhavan N. Rajeevan[6]
[1]Indian Institute of Tropical Meteorology, Ministry of Earth Sciences, India
[2]National Center for Atmospheric Research, Boulder, CO, USA
[3]India Meteorological Department, Ministry of Earth Sciences, New Delhi, India
[4]Centre for Development of Advanced Computing (C-DAC), Pune, Maharashtra, India
[5]Centre for Atmospheric and Oceanic Sciences, Indian Institute of Science, Bengaluru 560 012, India
[6]Ministry of Earth Sciences, Prithvi Bhavan, Lodhi Road, New Delhi 110003, India
**Correspondence:** chinmayjena@tropmet.res.in, sachinghude@tropmet.res.in
**Abstract**
Elevated levels of fine particulate matter (PM$_{2.5}$) during winter-time have become one of the most important
environmental concerns over the Indo-Gangetic Plain (IGP) region of India, and particularly for Delhi. Accurate
reconstruction of PM$_{2.5}$, its optical properties, and dominant chemical components over this region is essential to
evaluate the performance of the air quality models. In this study, we investigated the effect of three different
aerosol mechanisms coupled with gas-phase chemical schemes on simulated PM$_{2.5}$ mass concentration in Delhi
using the Weather Research and Forecasting model with the Chemistry module (WRF-Chem). The model was
employed to cover the entire northern region of India at 10 km horizontal spacing. Results were compared with
comprehensive filed data set on dominant PM$_{2.5}$ chemical compounds from the Winter Fog Experiment
(WiFEX) at Delhi, and surface PM$_{2.5}$ observations in Delhi (17 sites), Punjab (3 sites), Haryana (4 sites), Uttar
Pradesh (7 sites) and Rajasthan (17 sites). The Model for Ozone and related Chemical Tracers (MOZART-4)
gas-phase chemical mechanism coupled with the Goddard Chemistry Aerosol Radiation and Transport
(GOCART) aerosol scheme (MOZART-GOCART) were selected in the first experiment as it is currently
employed in the operational air quality forecasting system of Ministry of Earth Sciences (MoES), Government
of India. Other two simulations were performed with the MOZART-4 gas-phase chemical mechanism coupled
with the Model for Simulating Aerosol Interactions and Chemistry (MOZART-MOSAIC), and Carbon Bond 5
(CB-05) gas-phase mechanism couple with the Modal Aerosol Dynamics Model for Europe/Secondary Organic
Aerosol Model (CB05-MADE/SORGAM) aerosol mechanisms. The evaluation demonstrated that chemical
mechanisms affect the evolution of gas-phase precursors and aerosol processes, which in turn affect the optical
depth and overall performance of the model for PM$_{2.5}$. All the three coupled schemes, MOZART-GOCART,
MOZART-MOSAIC, and CB05-MADE/SORGAM, underestimate the observed concentrations of major
aerosol composition (NO$_3^-$, SO$_4^{2-}$, Cl$^-$, BC, OC, and NH$_4^+$) and precursor gases (HNO$_3$, NH$_3$, SO$_2$, NO$_2$, and O$_3$)
over Delhi. Comparison with observations suggests that the simulations using MOZART-4 gas-phase chemical
mechanism with MOSAIC aerosol performed better in simulating aerosols over Delhi and its optical depth over
the IGP. The lowest NMB (-18.8%, MB = -27.4 µg/m$^3$) appeared for the simulations using MOZART-MOSAIC



scheme, whereas the NMB was observed 32.5% (MB = -47.5 µg/m$^3$) for CB05-MADE/SORGAM and -53.3%
(MB = -78 µg/m$^3$) for MOZART-GOCART scheme.



## 1. Introduction

The industrial activities in India have escalated to new heights in the past three decades which consequently have led to multiple urban environmental issues, especially deteriorating air quality due to suspended particulate matter of aerodynamic diameter smaller than 2.5 μm ($PM_{2.5}$) (Ghude et al., 2016; Ghude et al., 2020). Therefore, it has become a matter of serious concern for public health in India. Currently, the air quality in India, especially in the northern region of India in general and Delhi in particular, is among the poorest in the world (World Health Organization, 2018). Therefore, managing air quality levels in this region of India has emerged as a complicated task.

Recent studies indicate that the exposure to the exceptionally high level of outdoor $PM_{2.5}$ pollution in the National Capital Region (NCR) Delhi poses a serious health risk to the public in Delhi (Ghude et al., 2016; Guttikunda et al., 2013), particularly during the winter season. Diversity of emission sources (Chandra et al., 2018; Hakkim et al., 2019), larger use of fossil fuel such as transport, industries, etc. (Chen et al., 2020), and large scale intense open crop-residue burning in surrounding regions of Delhi (Jena et al., 2015a; Beig et al., 2020; Kulkarni et al., 2020) is responsible for extreme air pollution episodes in the NCR region under favourable meteorological conditions (Vadrevu et al., 2011; Gargava et al., 2015; Tiwari et al., 2018; Liu et al., 2018; Krishna et al., 2019; Chate et al., 2013; Beig et al., 2013; Parkhi et al., 2016). This has drawn significant academic and research interest in predicting high $PM_{2.5}$ levels using numerical air quality models (Guttikunda et al., 2012; Beig et al., 2013; Krishna et al., 2019; Ghude et al., 2020; Kulkarni et al., 2020). Few recent studies have tested the performance of air quality models, particularly WRF-Chem, in simulating hourly $PM_{2.5}$ concentrations in Delhi (Ojha et al.,2020; Chen et al., 2020; Ghude et al., 2020; Kulkarni et al., 2020). These studies suggested that simulating and predicting extreme air quality episodes, particularly $PM_{2.5}$ concentrations exceeding 300μg/m³, in the NCR region is a challenging task for the air quality models (Kumar et al., 2015; Krishna et al., 2019; Bali et al., 2019). Large uncertainties are involved in the prediction of atmospheric aerosols. This is because chemical transport models predictions suffer from errors in emission inventories (Jena et al., 2015b), inadequate understanding of some of the processes (e.g., secondary organic aerosol formation) (Balzarini et al., 2015), inaccuracies in the initialization of chemical and physical atmospheric state (Ghude et al., 2020), systematic and random errors because of numerical approximations, and approaches the different chemical mechanisms use to calculate size distribution of aerosols coupled with the gas-phase chemical mechanism.

A recent study showed significant differences in simulating aerosol mass concentration over China (Chen et al., 2016, 2017), Europe (Solazzo et al., 2012; Balzarini et al. 2015; Georgiou et al. 2018), USA (Yahya et al., 2017; Hodzic et al., 2013) and Tibetan Plateau (Yang et al., 2018). Differences in the chemical mechanism (Knote et al., 2015), parameterization of heterogeneous formation of secondary inorganic aerosols (SIA), and secondary organic aerosols (SOA), which affects the aerosol process and the evolution of gas-phase precursors, are found to play a key role in the reconstruction of aerosols in above studies. For organic aerosols, the complexity of secondary formation and its aging processes and the lack of emission estimates of intermediate-volatility and semi-volatile organic compounds affect the model performance (Chen et al., 2017; Tsigaridis et al., 2014). Balzarini et al. (2015) showed that simulated total PM mass concentrations, as well as



aerosol subcomponents, vary between the RADM2 gas-phase chemical mechanism with Modal Aerosol
Dynamics Model for Europe/Secondary Organic Aerosol Model (MADE/SORGAM) and CBMZ gaseous
parameterization with Model for Simulating Aerosol Interactions and Chemistry (MOSAIC) aerosol
mechanisms, and CBMZ-MOSAIC performed better in reproducing lower aerosol concentration than RADM2-
MADE-SORGAM. Yang et al., (2018) also reported that RADM2-MADE/SORGAM could simulate higher
surface $PM_{2.5}$ mass concentrations better than the CBMZ-MOSAIC module over the Tibetan Plateau because of
the difference in aerosol compounds and distribution of computed aerosol concentrations between modes and
bins. On the other hand, Georgiou et al. (2018) showed that simulated $PM_{2.5}$ by the RADM2-MADE/SORGAM
mechanism exhibit lowest mean bias when compared to observations, but it overestimates the ammonium and
sulfate aerosols. On the other hand, the MOSAIC aerosol mechanism overestimates $PM_{2.5}$ mass concentrations
substantially over the eastern Mediterranean region. In a recent study, Curi et al., (2015) showed that magnitude
of the uncertinities in AOD arrising from the assumations of aerosol mixing state (external, internal
homogeneous, and internal core shell), the chemical species density, the species complex refractive index, and
the hygroscopic growth factors is significant if compared with typical differences found in comparison of
simulated values with AOD observations.
Most of these studies focused over the USA, Europe, or China. However, a detailed evaluation of $PM_{2.5}$
with different coupled chemcial schemes (gas-phase mechanism with aerosol schemes) over the IGP region in
general and Delhi in particular with scare datasets left unclear view of WRF-Chem's ability to predict $PM_{2.5}$
over this region, a region documented to be one of the most polluted regions in South Asia. A very limited
number of modelling studies have focused on evaluating the performance of the air quality models in simulating
$PM_{2.5}$ mass concentration in Delhi on an hourly time scale during winter-time pollution. For example, Krishna et
al., (2019); Ghude et al., (2020); Kulkarni et al., (2020) carried out WRF-Chem simulations over Delhi with
MOZART-4 gas-phase chemistry and Goddard Chemistry Aerosol Radiation and Transport (GOCART) aerosol
mechanism. They found that the model very well captures the temporal variation in $PM_{2.5}$ mass concentration
driven by synoptic-scale meteorological variability, but shows substantial error in simulating the $PM_{2.5}$
magnitude and large model-observations differences. It is therefore important to evaluate the model capability in
simulating the concentration of major $PM_{2.5}$ components and major oxidants and how different chemical
mechanism affects the $PM_{2.5}$ mass concentrations over this region.
In this study, three two-month simulation experiments using Weather Research and Forecasting model
with chemistry (WRF-Chem v3.9.1) were designed for the Northern region of India in general, and National
Capital Region, Delhi in particular at 10 km grid resolution during winter-time. For this, we employ and inter-
compare MOZART-GOCART, MOZART-MOSAIC, CB05-MADE/SORGAM coupled gas-phase chemistry
and aerosol mechanisms to evaluate the simulated $PM_{2.5}$ mass concentrations with extensive ground-based
observations in Delhi (17 sites), Punjab (3 sites), Haryana (4 sites), Uttar Pradesh (7 sites), and Rajasthan (17
sites). We also investigated the optical properties of aerosols, and ability of the different coupled chemical
mechanism in the model to reconstruct the different aerosol components of $PM_{2.5}$ in Delhi using chemical
speciation observations from the Winter Fog Experiment (WiFEX) that took place at the Indira-Gandhi
International Airport, New Delhi (Ghude et al., 2017; Acharja et al., 2020). The comparison among the coupled



aerosol schemes aims at identifying the reasons for differences in model performance. Section 2 briefly
describes the gas-phase chemistry and aerosol mechanisms used for the simulations, the basic model
configuration, emission data used, and data used for the model evaluation. In section 3, we present the results
from the sensitivity simulations and their evaluation with surface observations. Our conclusions and suggestions
for further study are given in Section 4.

**2. Model setup and description**
In this study, we used the Weather Research and Forecasting model coupled with chemistry WRF-
Chem v3.9.1 to simulate surface $PM_{2.5}$ mass concentration during the peak winter period, starting from 1
December 2017 to 31 January 2018. Recently, the model has been widely used to simulate the air quality in
Delhi (Beig et al., 2013; Gupta and Mohan 2015; Ghude et al., 2020; Kulkarni et al., 2020; Chen et al., 2020)
and to estimate $NO_X$ and $PM_{2.5}$ mass concentration over India (Ghude et al., 2013; Krishana et al., 2019; Ojha et
al., 2020; Beig et al., 2020). In this study, three sets of simulations were designed using following three widely
used coupled  schemes (gas-phase chemical mechanisms with aerosol schemes) to simulate the $PM_{2.5}$ mass
concentrations over the northern region of India.
**MOZART-GOCART (MG):** In the first experiment, simulation is performed with the Model for Ozone and
related Chemical Tracers (MOZART-4) gas-phase chemical mechanism (Emmons et al. 2010) coupled with the
Goddard Chemistry Aerosol Radiation and Transport (GOCART) aerosol scheme (Chin et al., 2000; Ginoux et
al., 2001). It includes 157 gas-phase reactions, 85 gas-phase species, 39 photolysis, and 16 bulk aerosol
compounds. For this experiment, the chemistry scheme is consistent with the chemistry used in the global model
that provides the chemical initial and boundary conditions. The GOCART aerosol model simulates five major
types of aerosols, namely, sulfate, black carbon, organic carbon, dust, and sea salt. GOCART scheme does not
simulate nitrate and secondary organic aerosols. The composition of GOCART aerosol module includes fine
unspeciated aerosol contribution ($P_{25}$), organic carbon (hydrophobic OC1 and hydrophilic OC2), organic black
carbon (hydrophobic BC1 and hydrophilic BC2), sulfate ($SO_4^{2-}$), dust of different sizes ($D_1$, $D_2$, $D_3$, $D_4$ and $D_5$
representing dust with effective radii of 0.5, 1.4, 2.4, 4.5 and 8 μm respectively),  and sea salt of different sizes
($S_1$, $S_2$, $S_3$ and $S_4$ representing sea salt with effective radii of 0.3, 1.0, 3.25 and 5μm respectively).
**MOZART-MOSAIC (MM):** In the second experiment, we used MOZART-4 gas-phase chemical mechanism
coupled with the Model for Simulating Aerosol Interactions and Chemistry (MOSAIC) (Zaveri et al., 2008)
aerosol scheme. MOSIAC includes sulfate (SULF = $SO_4^{2-}$ +$HSO_4^-$), methanesulfonate ($CH_3SO_3$), ammonium
($NH_4^+$), sodium (Na), calcium (Ca), nitrate ($NO_3^-$), chloride ($Cl^-$), carbonate ($CO_3$), black carbon (BC), primary
and organic mass (OC). Other unspecified inorganic species, inert minerals, and trace metals are lumped
together as OIN (other inorganic mass). MOSIAC also allowed the gas-phase to partition to the particle-phase,
which include $H_2SO_4$, $HNO_3$, HCl, $NH_3$, and MSA (methanesulfonicacid), and also include secondary organic
aerosols (SOA). MOSAIC uses a sectional aerosol bin approach for the representation of the aerosol size
distribution. In the WRF-Chem model, one can choose between four and eight aerosol size bins, which are
demarcated by their lower and upper dry particle diameters. In both cases, only one bin is assigned to aerosols
with a diameter between 2.5 and 10 μm. Therefore, when four aerosol bins are used, three bins are assigned to





160 aerosols less than 2.5 µm in diameter. When eight aerosol bins are used, seven bins are assigned to aerosols with

161 diameters within this range. Usually, it is sufficient to use the four-bin simulation option to which the focus is

162 on air quality and it also reduces computational complexity (Georgiou et al., 2018).

163 **CB05-MADE/SORGAM (CMS):** In the third experiment, we conducted simulations using the Carbon Bond 5

164 (CB-05) gas-phase mechanism (Yarwood et al., 2005,) which includes 51 chemical species and 156 reactions.

165 Aerosol processes are represented by the Modal Aerosol Dynamics for Europe/ Secondary Organic Aerosol

166 Model (MADE/SORGAM) (Ahmadov et al., 2012) which uses modal aerosol size distribution, and includes an

167 advanced secondary organic aerosol (SOA) treatment based on gas-particle partitioning and gas-phase oxidation

168 in volatility bins. The CB05-MADE/SORGAM mechanism has also been coupled to existing model treatments

169 of various feedback processes such as the aerosol semi-direct effect on photolysis rates of major gases and the

170 aerosol indirect effect on cloud droplet number concentration and resulting impacts on shortwave radiation

171 (Yahya et al., 2016).

172   The model domain covers the entire northern region of India at a horizontal grid spacing of 10 km and

173 47 vertical levels stretching from the surface to 10 hPa. Prior anthropogenic emissions of aerosols and trace

174 gases ($PM_{2.5}$, $PM_{10}$, OC, BC, CO, NOx, etc.) were taken from the EDGAR-HTAP (Emission Database for

175 Global Atmos. Res. for Hemispheric Transport of Air Pollution) for the year 2010 at 0.1° x 0.1° grid resolution

176 and scaled to 2018 using scaling factors as given in Venkatraman et al. (2018). No diurnal variation was added

177 to emissions. Biogenic emissions are calculated online using the Model of Emissions of Gases and Aerosols

178 from Nature version 2.1 (MEGAN2.1) (Guenther et al., 2006) and dust emissions are based on the online

179 Atmospheric and Environmental Research Inc. and Air Force Weather Agency (AER/AFWA) scheme (Jones

180 and Creighton, 2011). Emissions from sea salt are generated based on the scheme of Gong et al. (1997). Daily

181 open biomass burning emissions are obtained from the Fire INventory from NCAR (FINNv1.5)

182 (http://bai.acom. ucar.edu/Data/fire/). The chemical initial and lateral boundary conditions come from the global

183 model simulations from the Model for Ozone and Related Chemical Tracers (MOZART-4) and the

184 meteorological initial and lateral boundary conditions are provided by National Centers for Environmental

185 Prediction Final Reanalysis (NCEP/FNL) dataset, which is available every 6 hours. The simulations are

186 reinitialized monthly to constrain meteorological fields toward NCEP/FNL reanalysis data while forwarding

187 chemistry fields from the previous day. The details configuration of physics and chemistry options used in this

188 study, as well as their corresponding references, can be found in Table S1.

189 **2.1. Observational datasets and evaluation protocol**

190   The surface $PM_{2.5}$ data used in this study are taken from the air quality monitoring network operated by

191 the Indian Institute of Tropical Meteorology (IITM) and the Central Pollution Control Board (CPCB) in Delhi

192 (17 sites), and CPCB monitoring network in Punjab (3 sites), Haryana (4 sites), Uttar Pradesh (7 sites), and

193 Rajasthan (9 sites). The details of these monitoring locations are given in Table S2, and the geographical

194 locations are shown in Figure 1. These sites are representative of traffic, airport, urban, and suburban areas. The

195 quality control and assurance method, followed by CPCB for these air quality monitoring stations, is given at

196 https://cpcb.nic.in/quality-assurance-quality-control/. Furthermore, we take the following steps to reassure the

197 quality of $PM_{2.5}$ observations from the CPCB network stations. For Delhi data quality, we rejected all the



observations values below 10 µg/m$^3$ and above 1500 µg/m$^3$ at a given site if other sites in the network do not show values outside this range. The purpose of this step is to eliminate any short-term local influence that cannot be captured in the models and to retain the regional-scale variability. Second, we removed single peaks that are characterized by a change of more than 200 µg/m$^3$ in just one hour for all the data in CPCB monitoring stations. This step filters random fluctuations in the observations. Third, we removed some very high PM$_{2.5}$ values that appeared in the time series right after the missing values. For any given day, we removed the sites from the consideration that either experience instrument malfunction and/or appear to be very heavily influenced by strong local sources. Measurement of the inorganic aerosol composition (chemical ions) CL$^-$, NO$_3^-$, SO$_4^{2-}$, and NH$_4^+$ are made using MARGA-2S instrument during 01 December 2017 to 31 January 2018 at Delhi as a part of the WiFEX field campaign at Delhi international airport (Ghude et al., 2017; Acharja et al., 2020). The quality assurance and control process applied to the measurement of the chemical ion is given at Acharja et al., (2020). The meteorological observation data used in this study are taken from the Indian Meteorological Department (IMD).

The focus of the model evaluation was mainly to assess whether the model is able to effectively reproduce the spatial and temporal distributions of ambient total PM$_{2.5}$ mass concentrations and key PM$_{2.5}$ aerosol composition in Delhi as compared to observations. WRF-Chem is currently employed in the operational air quality forecasting system of the Ministry of Earth Sciences (MoES), Government of India. It is therefore important to examine the performance assessment of WRF-Chem for air quality simulations on a regional scale in general and over Delhi in particular during heavy winter-time pollution. Statistical evaluation metrics such as mean bias (MB), Pearson's correlation coefficient (R), normalized mean bias (NMB), normalized mean error (NME) (the definition of those measures can be found in Yu et al., 2006, and Zhang et al., 2006), and index of agreement (IOA) ranging from 0 to 1 (Yahya et al., 2016) with a value of 1 indicating a perfect agreement, is used to evaluate the perforation of different sets of the experiment. For evaluation, the observational data are paired up with the simulated data on an hourly basis for each site, and then observational data and simulated data are averaged out for all sites in Delhi, Haryana, Uttar Pradesh, and Rajasthan. The statistics are then calculated based on the state-specific data pairs.

## 3. Results and discussion

### 3.1. Meteorological evaluation

To quantitatively evaluate the model performance for basic meteorological parameters, the data for the temperature at 2m (T$_{2m}$), relative humidity at 2m (RH$_{2m}$), and wind speed at 10m (WS$_{10m}$) from six stations over Delhi, India is used. Statistical metrics are derived by comparing the output of the three model simulations to hourly measurements averaged over all ground stations. Table 1 shows the correlation coefficient (r), mean bias (MB), and root mean squared error (RMSE) between observed and modeled temperature at 2m (T$_{2m}$), relative humidity at 2m (RH$_{2m}$), and wind speed at 10m (WS$_{10m}$) over Delhi, India. Modelled T$_{2m}$ is in good agreement with observations (NMB = 2 to 5 %) but shows higher RSME values (8.84 to 8.92$^o$C) for all three mechanisms. The statistic for RH$_{2m}$ indicates that the model has dry bias during winter for all the three mechanisms and model over-predicts WS$_{10m}$ by an average of ~1.2 ms$^{-1}$ for all three mechanisms. The over prediction of wind



speed and poor correlation could be due to the poor representation of surface drag exerted by the unresolved
topography, other smaller-scale terrain features, and building morphology (Mar et al., 2016; Zhang et al., 2013).

**3.2. Sensitivity simulation of different aerosol scheme**
**3.2.1. Fine particulate matter (PM$_{2.5}$)**

Figure 2 shows the comparison for temporal variation between observed and the modeled hourly PM$_{2.5}$
mass concentrations form the sensitivity simulations with the three aerosol mechanisms from 1 December 2017
to 31 January 2018 over Delhi. Observed (black) surface PM$_{2.5}$ mass concentrations are averaged from the 17 air
quality monitoring stations in Delhi, while simulated PM$_{2.5}$ are for the MG (red), MM (blue), and CMS (green)
experiments are averaged from the 17 grids containing these observation sites. It can be seen that the run with
the MG, MM and CMS chemical schemes did not perform well, although it could capture the temporal variation
associated with the synoptic-scale variability during the study period. The mean observed PM$_{2.5}$ concentration
during peak winter months was about 191 µg/m$^3$. Whereas, the mean modeled PM$_{2.5}$ concentration vary from
89.9 µg/m$^3$ with the MG mechanism to 163.8 µg/m$^3$ and 147.1 µg/m$^3$ with the MM and CMS mechanism
respectively, showing a large variability in simulated PM$_{2.5}$ in Delhi among these mechanisms. All three
simulations with MG, MM and CMS chemcial schemes significantly under-predict observed PM$_{2.5}$
concentration averaged over Delhi. The statistic showed a large mean bias of about -101µg/m$^3$ (RMSE = 146.3)
for the simulation with the MG mechanism, which was about 53% of the corresponding observation. On the
other hand, simulations with the CMS and MM mechanisms showed much better agreement. The performance
statistic showed that the magnitude of bias decrease to -44 µg/m$^3$ (23%) and -27 µg/m$^3$ (14%) in the CMS and
MM simulations, respectively. Differences between the MG, MM, and CMS simulation are more pronounced
during the days when hourly PM$_{2.5}$ exceeds 250 µg/m$^3$, particularly on 1-7 and 25-31 December 2017, and 17-20
January 2018. Simulations with the MG mechanisms show poor ability of the model to capture these heavy
pollution days, while the latter two show reasonable ability to capture hourly PM$_{2.5}$ that exceeds 250 µg/m$^3$. On
some days, none of the simulation experiments captured the abrupt increase in PM$_{2.5}$ values observed on 18 -23
December, 26-28 December, and 8-16 January and underestimated the observed levels at the majority of the
stations.

Further, we evaluated the robustness of model performance for the individual stations in NCR Delhi
region. Table S3 shows the statistical performance of three MG, MM and CMS chemical schemes for seventeen
stations. Again, the MG mechanism showed the poorest performance with model mean bias varying from 19%
to 65% among different stations. The statistics show that for some stations, the MM mechanism performed quite
well, while the CMS mechanism shows better agreement for the others (Table S3). Surface PM$_{2.5}$ concentration
simulated with the MM mechanism show normalized mean bias (NMB) within ±15% for the following sites:
CRRI Mathura (-1.9%), ITO (-7.3%), Lodhi Road (-4.8%), North Campus DU (-12.6%), and Shadipur (-
11.5%). The sites showing the NMB within ±15% for the CMS mechanism are Burari Crossing (-11.51%),
CRRI Mathura (-7.3%), ITO (-12.1%), and Lodhi Road (-2.1%). Overall, the MM mechanism shows better
performance for simulating hourly PM$_{2.5}$ mass concentration over individual stations and Delhi as a whole, but


both the MM and CMS mechanisms show significant variability in NMB among the monitoring stations. This
could be because of the anthropogenic emissions of aerosols and trace gases taken from the EDGAR- HTAP at
0.1° x 0.1° grid resolution, which does not resolve the real variability in emissions in Delhi and may not
accurately capture true values observed at the point of measurement. Simulations with the MG mechanism
under-predicts $PM_{2.5}$ within 70% at all stations, possibly due to lack of $NO_3^-$ and secondary organic aerosols
(SOA) in the GOCART model. We find that simulated mean $NO_3^-$ and SOA together contributed ~44 µg/m³
with the MM mechanism, which is about 30% of total $PM_{2.5}$ mass concentration simulated during the winter
period.

We also examined the model performance of MG, MM and CMS chemical schemes over the Punjab,
Haryana, Uttar Pradesh, and Rajasthan (Figure 3), which are the neighboring states of Delhi and often influences
the air quality in NCR region (Kumar et al., 2015; Kulkarni et al., 2020). Table S4 shows the summary of the
performance statistic for the individual sites in each state. The average observed $PM_{2.5}$ concentration over
Punjab was 84.21 µg/m³ and WRF-Chem showed biases of about -24.7 µg/m³ (RMSE = 52.1), 13.1 µg/m³
(RMSE = 52.6) and 1.9 µg/m³ (RMSE = 48.1) for the MG, MM, and CMS aerosol mechanisms respectively.
This is about 29%, 15%, and 2% of the observed average value for the MG, MM, and CMS mechanisms,
respectively. For the individual monitoring stations in Punjab, simulations with the CMS mechanism showed
better performance with a bias of about 5.3% for Amritsar and 9.4% for Ludhiana, whereas the MM mechanism
showed better performance with a bias of about -4.8% for Gobindgarh RIMT  station.

The average observed $PM_{2.5}$ concentration over Haryana was about 138.8 µg/m³, significantly higher
(by 45%) than that of average $PM_{2.5}$ over Punjab. WRF-Chem over Haryana showed a bias of about -82.7 µg/m³
(-59.6%), -43.7 µg/m³ (-31.5%) and -58.9µg/m³(-42.4%) for the MG, MM and CMS chemcial mechanisms
respectively, indicating that all three aerosol mechanisms significantly under-predict $PM_{2.5}$ surface
concentration. For the individual monitoring stations in Haryana, simulations with the MM mechanism showed
better performance, and NMB is found to vary from -22% to 48% relative to observations. For Uttar Pradesh,
the MG mechanism showed the largest bias of about -126.5 µg/m³ (-62.2% of the observed value) while the MM
and CMS showed biases of about -58.4 µg/m³ (-28.5%) and -84.6 µg/m³ (-42.4%) respectively, indicating a
large error in simulations irrespective of the mechanism used. For the individual monitoring stations in Uttar
Pradesh, NMB with simulations with the MM mechanism found to vary from -21% to 63% relative to
observations. Relative to Haryana and Uttar Pradesh, performance statistics for Rajasthan show better results in
terms of bias for MM mechanism (bias = -7.6µg/m³, NMB = -8.1%). Other two chemicall mechanisms, MG and
CMS, showed biases of about -43.1 µg/m³ (-46%) and -22.3 µg/m³ (-24%), respectively. Overall, all three MG,
MM and CMS chemcial mechanisms tend to underpredict the observed $PM_{2.5}$ concentration over the majority of
stations in northern India, but  the MM mechanism was found to be performing better over Delhi and
neighbouring states, except Punjab, where the CMS mechanism performs the best.







**3.2.2. Comparison with satellite AOD**
We further examined how the differences between coupled chemical mechanisms translate in
simulating Aerosol Optical Depth (at 550 nm) over the model domain. Figure 4 shows the spatial distribution of
observed mean AOD (MODIS/TERRA) and simulated AOD at TERRA overpass time with three aerosol
mechanisms. All three mechanisms under-predict the observed AOD, and the difference between the three
mechanisms is more pronounced over the central and eastern regions of IGP. The mean AOD difference was the
highest (-58%) for the simulation with MG mechanism, while the latter two show reasonably good agreement
with a mean bias of about -4.3% and -6.6% for the MM and CMS mechanism. This indicates the crucial role of
the fine particle of aerosols, which have higher scattering efficiency, in aerosol optical depth budget (Seinfeld et
al., 2016; Balzarini et al. 2015; Yang et al., 2018). In spite of good agreement with mean AOD, simulations with
both the MM and CMS mechanisms show still large bias over the central and eastern regions of IGP compared
to other regions in the model domain. With the simulation with MG mechanism, the difference in magnitude
between observed and modelled AOD vary from -0.6 to -0.8 over this region. The observed differences between
simulated and observed AOD values over this region are consistent with results from previous studies (Kulkarni
et al., 2020 and Nair et al., 2012). These studies found that underestimation of anthropogenic emissions in the
IGP and errors in simulating dust emission and transport over this region are some of the reasons for differences
in observed and modelled AOD. However, given that emissions are constant in all the three simulation
experiments, the considerable differences between modelled and observed AOD might partially coming from
the difference in the simulation of the aerosol composition and dust scheme. In our simulation, the MG and
CMS use GOCART/AFWA dust scheme while MM uses the GOCART dust scheme.  Some of the previous
studies have shown that ammonium sulfate ($(NH_4)_2SO_4$), ammonium bisulfate ($(NH_4)HSO_4$), ammonium nitrate
($NH_4NO_3$) and ammonium chloride ($NH_4Cl$) scatter light more efficiently at 550 nm (Seinfeld et al., 2016),
while BC absorbs the light at 550 nm. The spatial pattern of mean BC concentration and concentrations of gas-
phase compounds that lead to secondary inorganic aerosols and distribution of secondary aerosols for the three
aerosol mechanisms is shown in Figure 5. We can see that the mean $SO_4^{2-}$ (Figure 5h) was generally lower in the
MG experiment and highest in the CMS experiment, particularly over the IGP and northeastern India. This
discrepancy may be related to less chemical aqueous-phase oxidation of $SO_2$ by $H_2O_2$ in MOZART-4 gas-phase
scheme because all the experiment shares the same $SO_2$ emissions. $H_2O_2$ is an efficient oxidant of sulphuric
compounds in clouds and fog. During peak winter months, widespread fog is often detected over the IGP region
during early morning hours and persists till late early afternoon (Ghude et al., 2017; Jenamani et al., 2015). As
shown in figure 5f, simulations with the MG and MM (MOZART-4 gas-phase chemistry) mechanism showed a
higher concentration of $H_2O_2$ over IGP, suggesting inefficient oxidation of $SO_2$ compared to the CMS
experiment. Figure 5i shows the surface $NO_3$ concentration simulated by the MM and CMS mechanism. Since
the MG mechanism does not simulate nitrate aerosols, $NO_3^-$ from the MG epxeriment is not shown here. Mean
$NO_3^-$ concentration was generally higher in the MM experiment than in the CMS experiment, particularly the
magnitude of $NO_3^-$ over central and eastern IGP region is larger. Similarly, as shown in Figure 5j, the magnitude
of mean $NH_4^+$ concentration was also higher in the MM experiment over central and eastern IGP. On the other
hand, mean $HNO_3$ concentration was found highest in the MG experiment, followed by the CMS experiment,
and the lowest was found in MM (Figure 5g) experiment. The highest $HNO_3$ concentration observed in the MG
experiment is related to the efficient photochemical conversion of $NO_2$ and OH to gas-phase $HNO_3$. However,



the lack of aerosol thermodynamics in the MG mechanism means that $HNO_3$ stays in the gas-phase and does not
partition to particle-phase. The main precursor for $NO_3$ is $HNO_3$, and the equilibrium between nitrate and $HNO_3$,
and gas-phase $NH_3$ and $HNO_3$ can convert to aerosol $NH_4NO_3$. This indicates that the gas-particle partitioning
from $HNO_3$ to $NH_4NO_3$ is more efficient in the MM experiment than in the CMS experiment. While, higher
$HNO_3$ concentration in the CMS experiment than in the MM experiment may be related to higher surface $NO_2$
(Figure 5b) concentration due to an efficient $O_3$-NO titration process that readily transforms to $HNO_3$ with the
photochemical reaction between $NO_2$ and OH, but not sufficiently converting to aerosol $NH_4NO_3$. The MM and
CMS experiment show higher BC concentration than the MG experiment (Figure 5d), but OC concentration is
higher in the MG experiment over the entire IGP region than the MM and CMS experiments (Figure 5e).
Further simulated BC to OC ratios is higher in the MG experiment over the IGP, compared to the other two
experiments. Few recent studies have shown the significant concentration of chloride ions ($Cl^-$) in the IGP
region (Ghude et al., 2017) and correlation of $NH_4^+$ with $Cl^-$ implied that sizeable fraction $NH_4^+$ with $Cl^-$
occurred in $NH_4Cl$ molecular form (Ali et al., 2019) through a gas-phase reaction between HCL and $NH_3$ (Du et
al., 2010). The primary source of this chloride is winter-time biomass and trash burning that occurred
widespread over the IGP region, but emissions of chloride from these sources are not provided to the model, and
therefore, the MM and CMS simulations show negligible concentrations of $Cl^-$ over the IGP (Figure not shown)
region. The bias between observed and simulated AOD may partially be related to missing $NH_4Cl$ aerosols in
the simulations. Magnitude of the uncertinities in mdoel AOD also arrisie from the assumations of aerosol
mixing state (external, internal homogeneous, and internal coreeshell), the chemical species density, the species
complex refractive index, and the hygroscopic growth factors. Recent study show that uncertinities in mdoel
AOD due to above paramter is significant if compared with typical differences found in comparison of
simulated values with AOD observations (Curi et al., 2015).


### 374    3.2.3. Major PM$_{2.5}$ speciation

Table 2 shows that all three chemcial mechanisms underestimate $PM_{2.5}$ concentrations in Delhi. The
lowest NMB appears for the MM mechanism (NMB = -18.8%, MB = -27.4 $\mu g/m^3$), whereas the NMB for the
GM mechanism is -53.3% (MB = -78 $\mu g/m^3$) and -32.5% (MB = -47.5 $\mu g/m^3$) for the CMS mechanism. Box-
whiskers' plot of observed $PM_{2.5}$ mass concentrations at IGI airport and its comparison with simulated $PM_{2.5}$
mass concentrations for the different aerosol mechanisms is given in Figure6. For observations, the $25^{th}$ and $75^{th}$
percentile of $PM_{2.5}$ values were observed between 75 $\mu g/m^3$ and 190 $\mu g/m^3$, whereas $25^{th}$ and $75^{th}$ percentile of
$PM_{2.5}$ for the MM, CMS, and MG chemcial mechanisums was observed between 75 $\mu g/m^3$ and 150 $\mu g/m^3$, 60
$\mu g/m^3$ and 120 $\mu g/m^3$, and 50 $\mu g/m^3$ and 90 $\mu g/m^3$, respectively. Among the three sensitivity experiments, the
median value of $PM_{2.5}$ for MM (100 $\mu g/m^3$) simulation was found closer to observation (120 $\mu g/m^3$). Overall,
$PM_{2.5}$ mass concentration simulated with the MM mechanism was found to be in better agreement with
observations.
In order to understand the individual components of $PM_{2.5}$ chemical species and examine the difference
in behavior by the aerosol mechanism for Delhi, we examine separately the dominant aerosol species in $PM_{2.5}$



and gas-phase compounds that lead to secondary inorganic aerosols. Figure 6 presents the box-whiskers plot for
components of $PM_{2.5}$ from the observations and simulated by the model for the different coupled aeorsol
mechanisms at Delhi during the study period. It should be noted that nitrate is absent in GOCART; therefore,
ammonium and nitrate are not shown in Figure 6. The MG mechanism does not simulate $NH_4$ but multiplies
sulfate by 1.375 to account for $NH_4$ mass in total $PM_{2.5}$ mass concentration. Observations at Delhi during the
study period suggest that the ratio of $NH_4$ to sulfate is about 1.545 during the winter season, which is about 11%
higher. Further, simulated mean sulfate aerosols ($SO_4^{2-}$) concentration was largely underestimated (~ 40 - 60%)
by the model in all three experiments with bias raining from 9 $\mu g/m^3$ to 14 $\mu g/m^3$ compared to the observations
(Table 2). However, the gas-phase precursor ($SO_2$) of sulfate aerosol simulated by the model was found to be
slightly overestimated by about 4 - 5 ppb in all the simulations. This implies that chemical aqueous-phase
oxidation of $SO_2$ by $H_2O_2$ and a heterogeneous nucleation rate from sulphuric acid ($H_2SO_4$) is not efficiently
simulated by all three mechanisms over Delhi during the winter period. Nitrate and sulfate interact with each
other through thermodynamic equilibrium but depends upon the gas-phase ammonia concentrations. It can be
seen that for $NH_3$, the simulations with MM and MG mechanism slightly overestimate $NH_3$ by about 2 – 4 ppb,
respectively. On the other hand, simulations with CMS mechanism underestimate $NH_3$ by about 8 ppb for the
same ammonia emissions. However, ammonium aerosols are underestimated by both the simulations with CMS
(MB = -26.3 $\mu g/m^3$) and MM (MB = -24.8 $\mu g/m^3$) mechanisms compared to observations (~34 $\mu g/m^3$).
Simulated mean nitrate concentration was generally higher in the MM (MB = -7.6 $\mu g/m^3$, NMB = ~25%)
experiment than in the CMS (MB = -19.4 $\mu g/m^3$, NMB = ~62%) experiment compared to observation (28
$\mu g/m^3$) but both the experiment show nitrate is negatively biased. As discussed earlier, the main precursor for
$NO_3$ is $HNO_3$, and the equilibrium between nitrate, $HNO_3$, and gas-phase $NH_3$. It can be seen that simulations
with the MM and CMS mechanisms highly underestimate the $HNO_3$ concentration during the winter period.
Overall, the gas-particle partitioning from $HNO_3$ to ammonium nitrate is not efficient in the MM and CMS
chemical mechanism. The difference between underestimation of simulated ammonium and nitrate aerosols in
the MM and CMS experiments could be due to the differences in the different treatment of the gas-to-particle
partitioning from the nitric acid to ammonium nitrate as a function of humidity (Balzarini et al., 2015; Georgiou
et al., 2018). Simulations with the MM and CMS simulations highly underestimate chloride aerosol
concentrations (by about 0.1 $\mu g/m^3$) compared to observations (25 $\mu g/m^3$) due to the absence of anthropogenic
chloride emissions. This indicates considerable uncertainty in the representation of tropospheric chloride
emissions and chemistry that affects aerosol formation. Earlier studies have also shown significant enhancement
in anthropogenic chloride (chemical tracer for garbage, plastics, and tires burning) during the peak winter season
(Acharja et al., 2020; Ghude et al., 2017). During cold winter nights, open biomass burning occurs on the streets
and numerous residential localities in the IGP. In the cold winter conditions, people burn wood, leaf litter,
garbage, plastics, and tires, etc. as these are available almost free-of-cost as compared to clean energy sources
for which one needs to pay. Compared to observations, organic carbon and black carbon is underestimated by all
the three experiments. The lowest mean bias (~ -11 $\mu g/m^3$, ~45%) for BC is found for the simulations with MM
and CMS mechanism, while the simulation with MG mechanism show ~65% bias (~ -16 $\mu g/m^3$) with respect to
the observed BC concentration. For OC the lowest mean bias (~ -9 $\mu g/m^3$, ~35%) appears for the MG
mechanism, whereas mean bias was ~ -12 $\mu g/m^3$, (~45%) for MM mechanism, and ~ -15 $\mu g/m^3$ (~51%) for
CMS mechanism. Because all the three experiments use the same emission sources, discrepancies between the
MG, MM, and CMS experiments could be partially attributed to differences treatment of aerosols calculations
by the modal and sectional bin approach. The MOSAIC scheme in this study uses Zaveri et al. (2008) approach
to divide aerosols into four bins, whereas the CMS scheme use Mozurkewich (1993) approach to divide aerosols
into three modes.

**4. Conclusion**

In this study, we simulated atmospheric gases and aerosols using three WRF/Chem modelling

configuration to investigate the effect of coupled gas-phase chemistry and aerosol mechanisms on the
reproductions of aerosol concentrations and aerosol optical depth over the northern region of India for the winter
period. Simulated results were compared with the air quality observational data from 17 sites in Delhi, 4 sites in
Haryana, 7 sites in Uttar Pradesh, 9 sites in Rajasthan over North India. Further, the performance of MOZART-
GOCART (MG), MOZART-MOSAIC (MM), and CB05-MADE/SORGAM (CMS) coupled gas-phase
chemistry and aerosol mechanisms were investigated for Delhi for major $PM_{2.5}$ chemical spices observed during
WiFEX field campaign at IGI Airport, Delhi. Performance of the model for basic meteorological parameters
indicates that WRF-Chem could capture 2 m temperate very well but overestimate the wind speed by about 1.2
$ms^{-1}$ at Delhi and may be related to the limited representation of the topography by the model.

Overall, all three coupled chemcial mechanisms tend to underpredict the observed $PM_{2.5}$ concentration

over the majority of stations in northern India, but the MOZART-MOSAIC mechanism was found to be
performing better over Delhi and neighbouring states. Surface $PM_{2.5}$ concentration simulated by the MOZART-
MOZAIC and CB05-MADE/SORGAM chemical mechanism demonstrated relatively lower bias compared to
MOZART-GOCART chemical mechanism. The model sufficiently captured the spatial distribution of mean
AOD in all three simulations, but MOZART-GOCART highly underpredicts the observed AOD compared to
the other two chemcial mechanisms. This is partly due to the difference in aerosols compounds and particularly
missing nitrate and secondary organic aerosols from the MOZART-GOCART mechanism. MOZART-MOZAIC
and CB05-MADE/SORGAM mechanism underestimate ammonium, nitrate, sulfate, BC, and OC aerosol mass
concentrations, and anthropogenic chloride is completely missing from the simulation. These fine mode aerosols
scatter/absorbed light more efficiently at 550 nm (Seinfeld et al., 2016), and underestimation of these species in
simulations MOZART-MOSAIC and CB05-MADE/SORGAM mechanisms is partly related to observed-
modelled bias for surface $PM_{2.5}$ and AOD over the region. Observations in Delhi show a significant contribution
of chloride aerosols in SOA, and missing sources of anthropogenic chloride emission lead to large-bias between
model and observed chloride concentrations. This is found to be one of the contributing factors for observed
discrepancies between surface $PM_{2.5}$ and AOD in all three experiments over the northern region of India. In
summary, the result suggests considerable uncertainty in MOZART-GOCART, MOZART-MOSAIC, and
CB05-MADE/SORGAM chemistry in the representation of aerosol chemical species and chemistry that affects
the aerosol formation. This further implies that the under-prediction of $PM_{2.5}$ concentrations in all three
chemical mechanisms is partially coming from the under-prediction of major aerosol species of fine particular
matter over IGI Airport, Delhi. Therefore, the selection of chemical mechanisms is a key aspect, and MOZART-



MOSAIC mechanism could perform better in reconstructing the AOD aerosols over the northern region of India
and surface PM$_{2.5}$ over Delhi and neighbouring states.

**Data availability**
The 0.1°× 0.1°emission grid maps can be downloaded from the EDGAR website
onhttps://edgar.jrc.ec.europa.eu/htap_v2/index.php?SECURE=_123 per year per sector. The model data is
available at Prithvi (IITM) super-computer and can be provided upon request to the corresponding author.
Observational data on PM$_{2.5}$ measurements can be obtained from the CPCB website on
https://app.cpcbccr.com/ccr.



**Author contributions**
All authors contributed to the research; CJ and SDG designed the research; CJ conducted the research; CJ and
SDG wrote the paper; CJ performed the WRF/Chem model simulations; RK  contributed to writing; SD, VKS,
PA, SHK, MK, AJK, DMC, KA, RN, and MR formulated the research.

**Competing interests**
The authors declare that they have no conflict of interest.

**Acknowledgment:**
We acknowledge the use of surface PM$_{2.5}$ data from air quality monitoring network of the Central Pollution
Control Board (CPCB), India. This work was supported by funding from the national monsoon mission project
of the Ministry of Earth Sciences (MoES). The use of MODIS data provided by NASA Earth Observing System
Data and Information System (EOSDIS). The emissions inventory for chemical species from the HTAP and
EDGAR emissions for 2010. We would like to acknowledge high-performance computing support from Aditya
and Pratyush provided by the Indian Institute of Tropical Meteorology, Pune. This work was supported by the
National Supercomputing Mission (NSM) program grant to the authors at C-DAC, and we are grateful to the
Executive Director and the Director-General of C-DAC.




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



**FIGURE CAPTIONS:**
**Figure 1:** Geographical locations of air quality monitoring stations. Delhi stations are represented by blue
circles; red circles represent stations in the states of Punjab, Haryana, Rajasthan, and Uttar Pradesh.
**Figure 2:** The time series of hourly $PM_{2.5}$ concentrations from the simulation of MOZART-GOCART (red),
MOZART-MOSAIC (blue), and CB05-MADE/SORGAM (green) aerosol mechanism was compared with the
air quality observation (black) form the CPCB data in Delhi region from 1 December 2017 to 31 January 2018.
**Figure 3:** The time series of hourly $PM_{2.5}$ concentrations from the simulation of MOZART-GOCART (red),
MOZART-MOSAIC (blue), and CB05-MADE/SORGAM (green) aerosol mechanism was compared with the
air quality observation (black) form CPCB data of Punjab, Haryana, Uttar Pradesh, and Rajasthan.
**Figure4:** Spatial distribution of simulated AOD of CB05-MADE/SORGAM, MOZART-MOSAIC, MOZART-
GOCART, and its difference with observed mean AOD from MODIS.
**Figure 5:** Spatial distribution of simulated surface $SO_2$, $NO_2$, Ozone, BC, OC, $H_2O_2$, $HNO_3$, $SO_4^{2-}$, $NO_3^-$, $NH_4^+$
of CB05-MADE/SORGAM, MOZART-MOSAIC, MOZART-GOCART model.
**Figure 6:** Box-whisker plots of Nitrate ($NO_3^-$), Ammonium ($NH_4^+$), Chlorine (Cl), Organic Carbon (OC), Black
Carbon (BC), Sulfate ($SO_4^{2-}$), $HNO_3$, $SO_2$, $NH_3$, $NO_2$, Ozone and $PM_{2.5}$ for the MOZART-GOCART (MG),
MOZART-MOSAIC (MM), and CB05-MADE/SORGAM (CMS) mechanisms over IGI Airport, Delhi.







**Table1**. Pearson's correlation coefficient (R), mean bias (MB), and root mean squared error (RMSE) of hourly values of temperature at 2m, relative humidity at 2m, planetary boundary layer height, and wind speed at 10m for the MOZART-GOCART (MG), MOZART-MOSAIC (MM), and CB05-MADE/SORGAM(CMS) mechanisms averaged over all stations in Delhi.

| State | Variables | MOZART-GOCART | | | MOZART-MOSAIC | | | CB05-MADE/SORGAM | | |
|---|---|---|---|---|---|---|---|---|---|---|
| | | MB | RMSE | R | MB | RMSE | R | MB | RMSE | R |
| Delhi | $T_{2m}$ | 0.78 | 8.92 | 0.23 | 0.39 | 8.84 | 0.24 | 0.31 | 8.91 | 0.23 |
| | RH | -36.6 | 41.8 | 0.21 | -36.4 | 41.6 | 0.20 | -36.5 | 41.7 | 0.17 |
| | Wind Speed ($WS_{10m}$) | 1.2 | 2.0 | 0.26 | 1.2 | 1.9 | 0.25 | 1.1 | 1.9 | 0.27 |

**Table2:** Index of Agreement (IOA), mean bias (MB), normalized mean bias (NMB), and root mean squared error (RMSE) of hourly values of $PM_{2.5}$ for the MOZART-GOCART (MG), MOZART-MOSAIC (MM), and CB05-MADE/SORGAM (CMS) mechanisms over IGI Airport, Delhi.

| State | Station | Variables | MOZART-GOCART | | | | MOZART-MOSAIC | | | | CB05-MADE/SORGAM | | | |
|---|---|---|---|---|---|---|---|---|---|---|---|---|---|---|
| | | | MB | NMB (%) | RMSE | IOA | MB | NMB (%) | RMSE | IOA | MB | NMB (%) | RMSE | IOA |
| Delhi | IGI Airport | $SO_4^{2-}$ | -13.8 | -66.7 | 21.2 | 0.36 | -13.9 | -66.9 | 19.9 | 0.43 | -8.7 | -42.3 | 18.9 | 0.48 |
| | | BC | -15.7 | -68.4 | 20.8 | 0.46 | -10.4 | -45.1 | 18.2 | 0.52 | -10.8 | -47.1 | 18.3 | 0.52 |
| | | OC | -9.2 | -35.3 | 13.1 | 0.51 | -11.7 | -44.8 | 15.3 | 0.46 | -13.4 | -51.3 | 16.5 | 0.44 |
| | | $NH_4^+$ | - | - | - | - | -24.8 | -73.8 | 30.2 | 0.44 | -26.3 | -78.1 | 31.5 | 0.43 |
| | | $NO_3^-$ | - | - | - | - | -7.6 | -25.8 | 18.8 | 0.45 | -19.4 | -62.4 | 24.7 | 0.41 |
| | | $CL^-$ | - | - | - | - | - | - | - | - | - | - | - | - |
| | | $PM_{2.5}$ | -77.9 | -53.3 | 102.7 | 0.53 | -27.4 | -18.8 | 81.2 | 0.69 | -47.5 | -32.5 | 86.5 | 0.66 |



**Figure 1:**

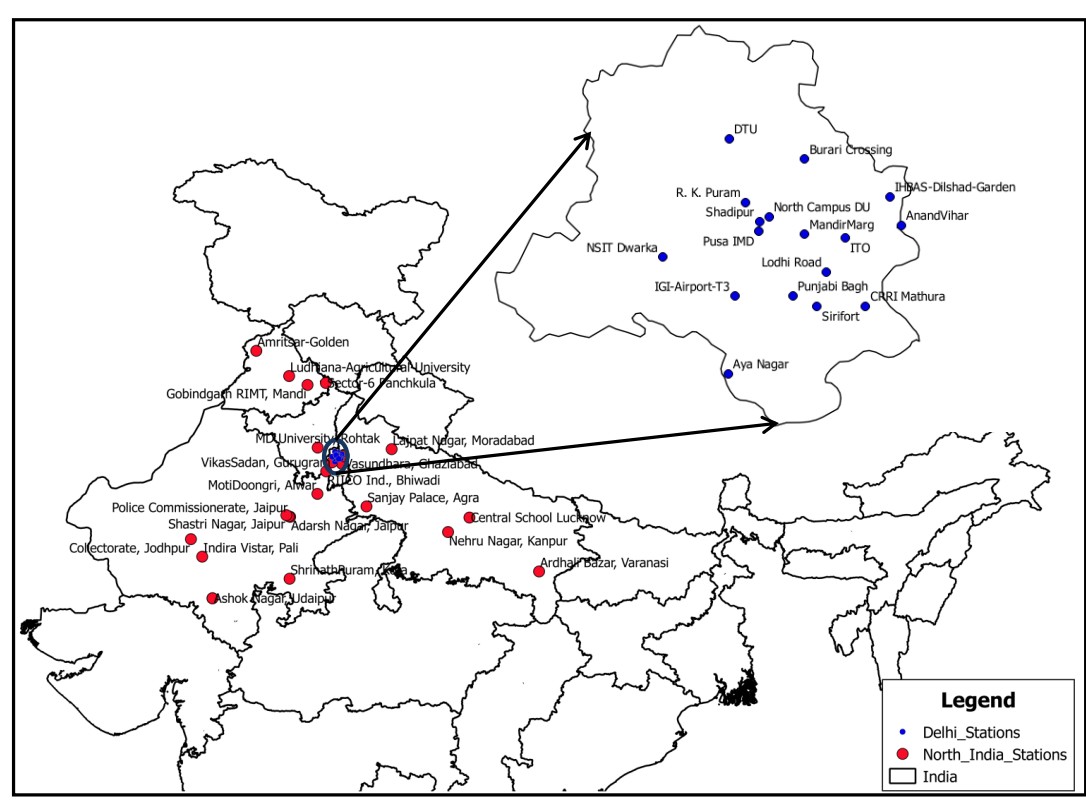



**Figure 2 :**

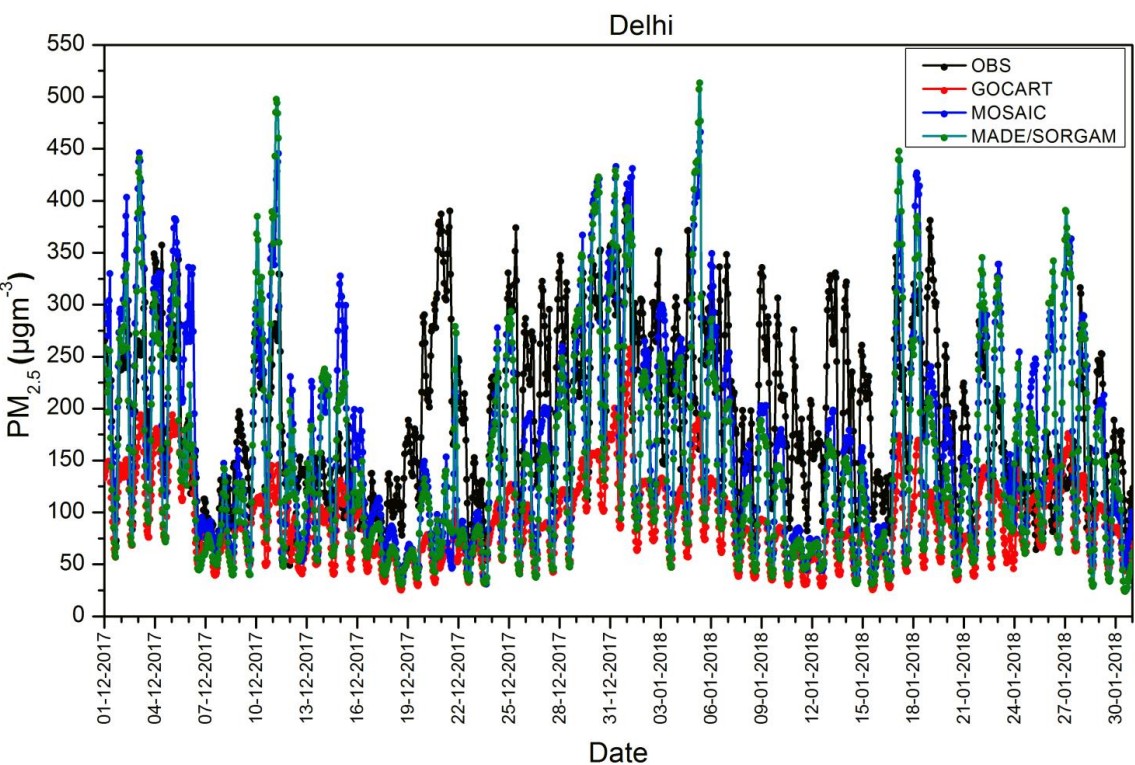



**Figure 3:**

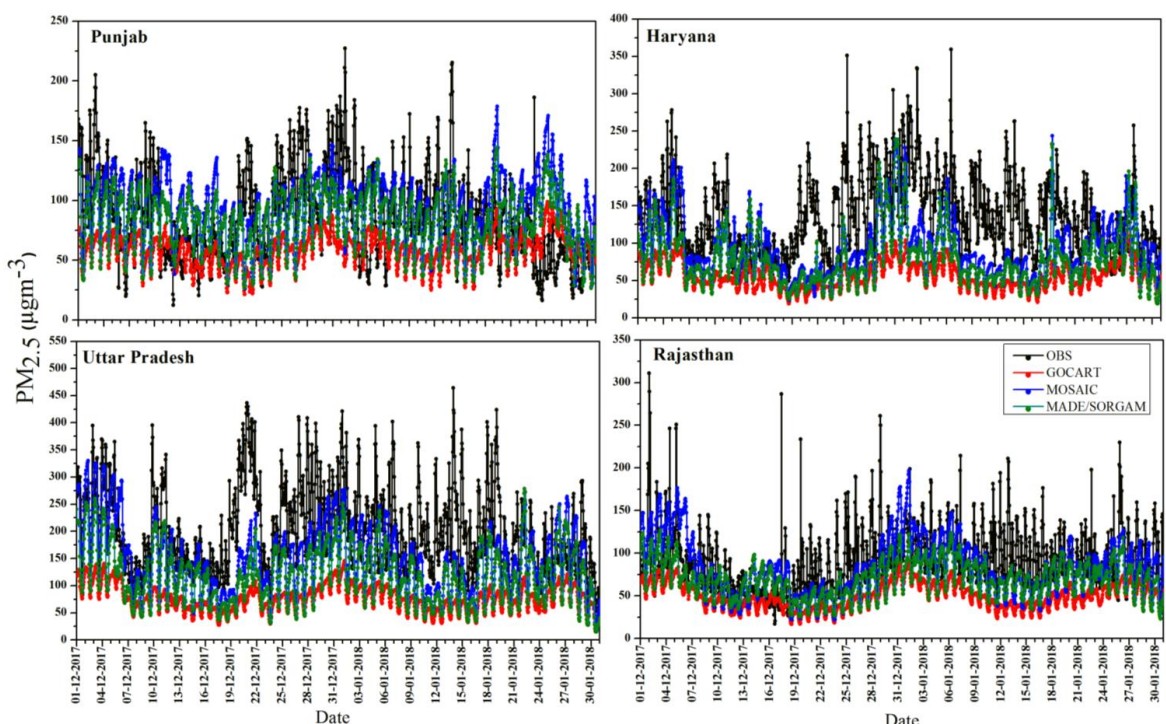





**Figure 4:**

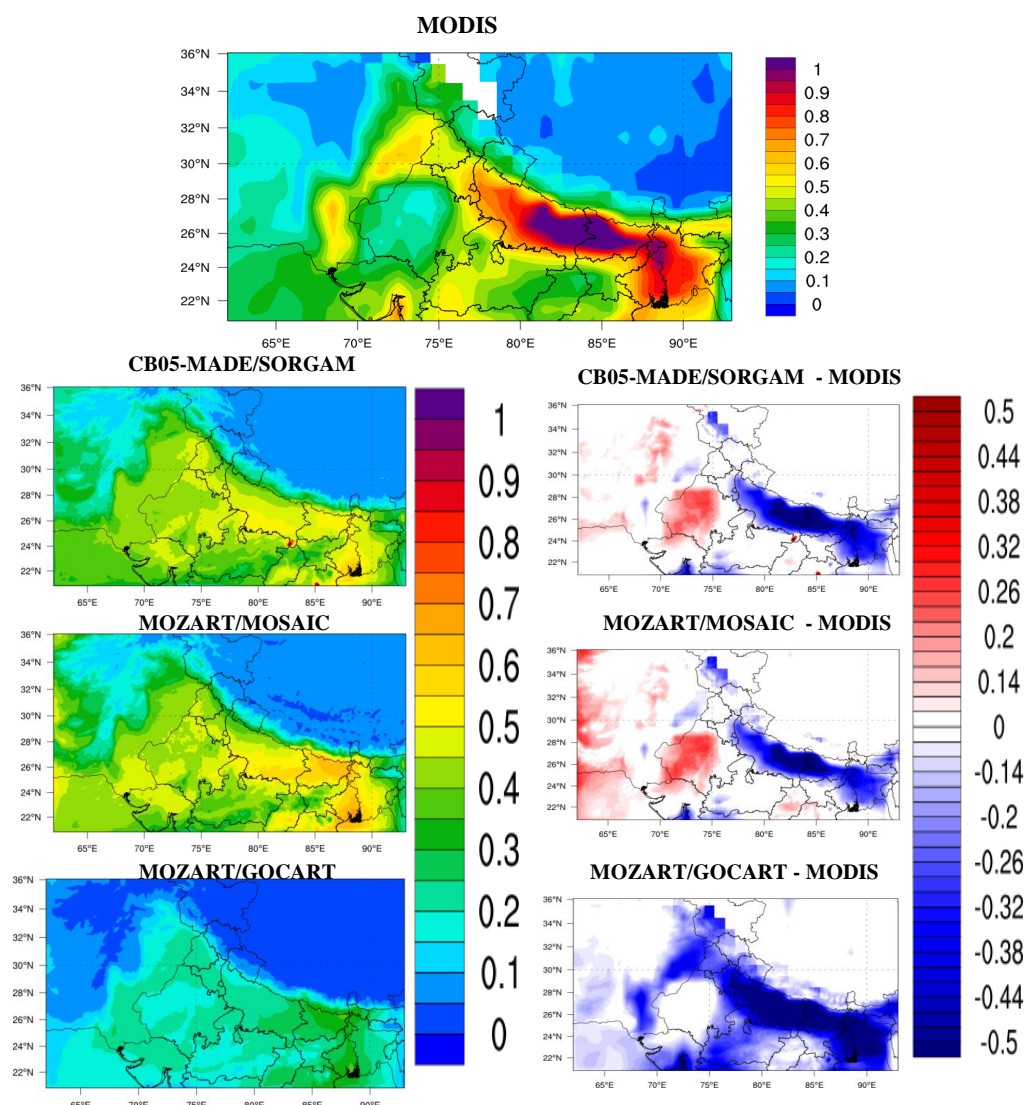



**Figure 5:**







**Figure 6:**

# IGI Airport T3, Delhi

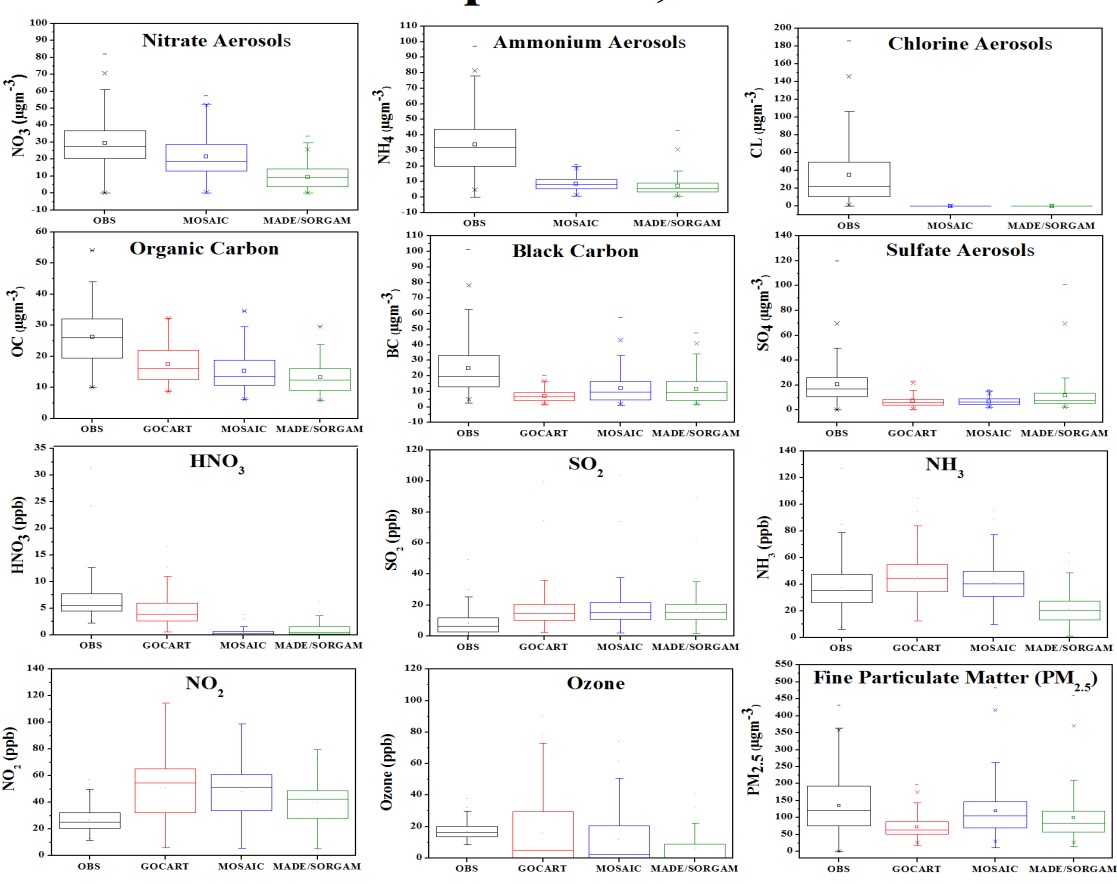