# Peer review of "Evaluating the sensitivity of fine particulate matter (PM2.5) simulations to chemical mechanism in Delhi"

_Atmospheric Chemistry and Physics, 2020_

## Referee Comment (RC1) · Anonymous Referee #1 · 11 Nov 2020

The authors have investigated the effects of three different aerosol mechanisms coupled with gas-phase chemical schemes on the simulated PM2.5 mass concentrations in Delhi using the WRF-Chem model. The manuscript is well written although some clarifications and improvements are necessary as outlined below before it can be recommended for publication.

**Comments**

1. It's not clear which SOA scheme was used in the MOZART-MOSAIC model configuration. Please specify with an appropriate reference.

2. How do the models perform with respect to dust? In the present manuscript no evaluation is presented for dust. It would be very useful to show modeled dust mass even if no speciated observations are available for dust.

3. The sentence at lines 410-411 doesn't make sense. For instance, the MM and CMS models appear to partition nearly all the available $HNO_3$ to $NH_4NO_3$ while some $NH_3$ still remains available in the gas phase. This indicates that $NH_4NO_3$ was limited by the formation of $HNO_3$. It does not mean that the models are inefficient in partitioning $HNO_3$ to $NH_4NO_3$. Since the models also predict higher $NO_2$ than observed, then it suggests that not enough $NO_2$ is oxidized to $HNO_3$ in the gas phase in both models. Can the authors comment on this aspect of the model? How are the models performing in $HNO_3$ production via $NO_2$ oxidation by OH radicals during the day and via $N_2O_5$ hydrolysis at night?

4. Since all models have the same emissions of $NH_3$, why does CMS predict lower NH3 and $NH_4^+$ than in the MM model (Figure 6)? $NH_3 + NH_4^+$ should be conserved. So if $NH_3$ is underpredicted than $NH_4^+$ should be overpredicted. But that doesn't seem to be the case here.

**Editorial Comments**

Line 18: Change "effect" to "effects".

Line 22: Change "filed" to "field".

Line 35: Replace "composition" with "species".

Line 100: Change "scare" to "scarce".

Line 151 and 154: MOSAIC is spelled incorrectly.

Line 161: Replace "option to which the focus" with "option when the focus". Also, clearly state which option was used in this study.

Line 205: Change "CL⁻" to "Cl⁻".

Line 206: Please spell out the MARGA acronym and provide a reference for the instrument's performance.

Line 327: Insert "be" after "might"

Line 330: Please remove the brackets around NH4 in ammonium bisulfate. It should be written as $NH_4HSO_4$.

Line 351: $NO_3$ should be $NO_3^-$

Line 351-352: Suggest changing this sentence to simply: "Particulate $NH_4NO_3$ is formed from condensation of gas-phase $NH_3$ and $HNO_3$.

Line 353: Delete the first word "While".

Line 362: Change "HCL" to "HCl".

Line 408: $NO_3$ should be $NO_3^-$

Figure 2: Suggest removing the black line from the observed profile and removing the filled circles from the simulated profiles. This should make the plots less congested and easier to read.

Figure 6. Is it OC plot showing organic carbon mass or is it actually organic aerosol mass?

Figure 6. The symbols in the gas-phase plots are barely visible, and please add a legend to explain all the symbols in the box plot.

Figure 6. Change "Chlorine Aerosols" to "Chloride". Additionally, change "Nitrate Aerosols" to "Nitrate", "Ammonium Aerosols" to "Ammonium", and "Sulfate Aerosols" to "Sulfate"

---

## Referee Comment (RC2) · Anonymous Referee #2 · 19 Nov 2020

In this paper, the authors report WRF-Chem simulations for the Indo-Gangetic Plan region of India, by focusing on Delhi for the Winter Fog Experiment during winter 2017-2018. The authors conducted coupled meteorology-chemistry simulations by deploying the state-of-the-art atmospheric chemistry model WRF-Chem. Three chemistry mechanisms with different levels of complexity were tested. All the simulations are evaluated by comparing the model output with the in-situ measurements of various meteorological parameters and chemical species, and the satellite AOD data. I have reservations with respect to the design of the numerical experiments and the interpretation of the model results.

Also, there are a number of operational global chemical weather models simulating air quality on a comparable spatial resolution ( $\sim$ 10km). Given the relatively poor performance of the regional model simulations in this study, it isn't clear what advantages can the presented model configurations offer over the existing global air quality forecast models. Therefore, the paper cannot be published in the present form due to the short-comings that are discussed below. I encourage the authors to substantially improve the quality of the model simulations and analysis of the results for future publications.

Model configuration: The lines 185-187 are confusing. Was the model simulated without reinitialization for a time period of a month? Since meteorological data assimilation or nudging isn't used here, monthly reinitialization would lead to a strong deviation of the model from the real weather.

I find the discussion of the meteorological evaluation insufficient. According to Table 1, the mean bias for the relative humidity is about -36%. This is a huge underestimate of the humidity, which also indicates that the model wasn't able to capture the fog events. For instance, this is necessary to simulate the sulfate formation in the cloud phase. Figure 6 shows that all the simulation cases overestimate the SO2 mixing ratios. On the other hand, the sulfate concentrations are underestimated at the Delhi airport site, indicating insufficient SO2 to sulfate production in the model.

Moreover, the RMSE for the RH and T2 are too high, while the correlations for all the meteorological parameters are very poor ( $R^2

the hourly variability of the anthropogenic emissions (especially the on-road component) is extremely important. The authors evaluate the simulated hourly PM2.5 mass concentrations by comparing them against the PM2.5 measurements. However, the ingested anthropogenic emissions don't have any hourly variability. While I understand that the EDGAR/HTAP emissions don't provide information about the hourly variability of the emissions, the authors could easily impose such variability in the emissions based on the traffic and other bottom-up information.

Another shortcoming of the anthropogenic emissions used in this study is the lack of day-to-day variability, e.g. weekdays vs. weekends. I assume the day-to-day variability of the emissions for Delhi is significant. The lack of both hourly and daily variabilities of the anthropogenic emissions can explain the large biases for most of the chemical species reported in the paper.

I assume all the anthropogenic emissions are added to the first model level. Therefore, the emissions from the power plants and other point sources aren't adequately included in the model, especially when the boundary layers are shallow. The implications of this approximation need to be discussed.

Photolysis: It isn't clear which photolysis schemes are used for the MOZART and CB-05 gas chemistry schemes. Some of the differences in the simulations of the secondary chemical species are caused by the differences in the photolysis schemes and also how they handle the cloud and aerosol feedback on the photolysis fluxes. The significant dry bias in the model can have a profound impact on the simulations of the photochemistry in the model.

Aerosol feedback: Aerosol feedback on the meteorology is included in all the model simulations. However, the discussion of such important processes in the model is very limited. It isn't clear how the different feedback processes (direct and indirect) are parameterized in the presented WRF-Chem simulations. Table 1 shows the temperature bias is quite different between the GOCART and the other two more advanced aerosol
schemes. I assume it's caused by the aerosol direct feedback. It'd greatly help to add a model case that doesn't include the aerosol feedback on the meteorology as a base case. It isn't clear how much the model performance improves by simulating the coupled meteorology-chemistry model over the model case without aerosol feedback. How about the aerosol indirect feedback? Its implications for simulating the fog events and so forth.

Fires: The role of the trash burning emissions in the urban areas are highlighted here. However, the FINN inventory doesn't include those emissions. This isn't discussed in the paper.

Lines 259-262: Are the observed spikes in the PM2.5 concentrations caused by shallow PBLH or an increase in the emissions compared to other days?

AOD evaluations: The dust species are included in the model simulations. There's a serious bug related to the dust AOD calculation code in WRF-Chem, which might explain the AOD underestimation due to the dust aerosols. It's reported in this publication: https://gmd.copernicus.org/preprints/gmd-2020-92/

I suggest adding a discussion about how the AOD is calculated in the model and potential uncertainties associated with this parameterization.

The primary goal of the paper is the selection of the optimal chemical mechanism for the Delhi region. As I noted above by improving the meteorological simulations, anthropogenic emission datasets and other components of the modeling system the air quality forecasting capabilities can be improved significantly regardless of the chemistry mechanism. Second, this study is limited to wintertime. These chemical mechanisms need to be tested for summertime also to select the optimal model configuration of WRF-Chem for the region.

Minor comments:

Include the domain plot with the terrain height.

**ACPD**
The name "MADE/SORGAM" is wrong. The SORGAM is a very old secondary organic aerosol scheme, whereas the chemistry scheme used here is based on the newer volatility basis set approach to simulate the SOA formation.

Figure 2-3: It's hard to distinguish different model cases in the time series plots.

Figure 4: What is the maximum MODIS AOD over the domain? It seems the AOD values >1 in some parts of the domain.